# Co-Existence of Oxazolidinone Resistance Genes *cfr*(D) and *optrA* on Two *Streptococcus parasuis* Isolates from Swine

**DOI:** 10.3390/antibiotics12050825

**Published:** 2023-04-28

**Authors:** Ning Han, Jie Li, Peng Wan, Yu Pan, Tiantian Xu, Wenguang Xiong, Zhenling Zeng

**Affiliations:** 1Guangdong Provincial Key Laboratory of Veterinary Pharmaceutics Development and Safety Evaluation, College of Veterinary Medicine, South China Agricultural University, Guangzhou 510642, China; 2National Laboratory of Safety Evaluation (Environmental Assessment) of Veterinary Drugs, College of Veterinary Medicine, South China Agricultural University, Guangzhou 510642, China; 3National Risk Assessment Laboratory for Antimicrobial Resistance of Animal Original Bacteria, College of Veterinary Medicine, South China Agricultural University, Guangzhou 510642, China

**Keywords:** *cfr*(D), *optrA*, oxazolidinones, resistance, *Streptococcus parasuis*

## Abstract

This study was performed to investigate the presence and characteristics of the oxazolidinone resistance genes *optrA* and *cfr*(D) in *Streptococcus parasuis*. In total, 36 *Streptococcus* isolates (30 *Streptococcus suis* isolates, 6 *Streptococcus parasuis* isolates) were collected from pig farms in China in 2020–2021, using PCR to determine the presence of *optrA* and *cfr*. Then, 2 of the 36 Streptococcus isolates were further processed as follows. Whole-genome sequencing and de novo assembly were employed to analyze the genetic environment of the *optrA* and *cfr*(D) genes. Conjugation and inverse PCR were employed to verify the transferability of *optrA* and *cfr*(D). The *optrA* and *cfr*(D) genes were identified in two *S. parasuis* strains named SS17 and SS20, respectively. The *optrA* of the two isolates was located on chromosomes invariably associated with the *araC* gene and Tn*554*, which carry the resistance genes *erm*(A) and *ant*(9). The two plasmids that carry *cfr*(D), pSS17 (7550 bp) and pSS20-1 (7550 bp) have 100% nucleotide sequence identity. The *cfr*(D) was flanked by GMP synthase and IS1202. The findings of this study extend the current knowledge of the genetic background of *optrA* and *cfr*(D) and indicate that Tn*554* and IS1202 may play an important role in the transmission of *optrA* and *cfr*(D), respectively.

## 1. Introduction

Oxazolidinones, such as linezolid and tedizolid, are regarded as the last-resort antibacterial compounds used to treat serious clinical infections caused by multidrug-resistant (MDR) Gram-positive bacteria, specifically methicillin-resistant strains of *Staphylococcus aureus* (MRSA) and vancomycin-resistant *Enterococci* (VRE) infections [1,2]. Statistics show that MRSA causes approximately 95,000 invasive infections and 19,000 deaths each year in the United States, which is a higher mortality rate than human immunodeficiency virus, viral hepatitis, tuberculosis and influenza combined [3,4]. Enterococci are of major importance in central line-associated bloodstream infections, catheter-associated urinary tract infections, ventilator-associated pneumonia and surgical site infections [5]. The target site of linezolid is the 50S large subunit of ribosomal proteins, especially the ribosomal proteins L3 and L4 that are encoded by the *rplC* and *rplD* genes, respectively [6]. However, with the widespread use of oxazolidinone antibiotics in livestock and poultry, many oxazolidinone-resistant bacteria have recently been reported [7,8,9,10]. The existence of transferable resistance genes, such as *optrA*, *poxtA*, *cfr*, and *cfr*-like genes, is considered to be one of the causes of oxazolidinone resistance [11,12].

The transferable oxazolidinone resistance genes, *optrA* and *cfr*, have been identified in considerable bacterial species worldwide. Since the *optrA* gene was first discovered in the *Enterococcus* spp. of human and animal origin in 2015 in China, numerous reports have indicated that the *optrA* genes exist in Gram-positive bacteria, such as *Enterococcus faecalis* and *Staphylococcus sciuri* [13]. The *optrA* gene confers transferable resistance to oxazolidinones and phenols by encoding an ATP-binging cassette (ABC-F) protein [14]. The *cfr* gene, initially isolated from *S. sciuri*, mainly confers multidrug resistance to lincosamides, oxazolidinones, phenols, and pleuromutilins by mediating methylation at position 2503 of the 23S rRNA gene [15]. The *cfr* and *cfr*-like genes (e.g., *cfr*(B), *cfr*(C) and *cfr*(D)) have been discovered in various Gram-positive and Gram-negative pathogens, such as *Staphylococcus*, *Enterococcus*, *Streptococcus suis*, *Escherichia coli*, and *Micrococcus caseolyticus* [16,17].

*Streptococcus parasuis*, once taxonomically classified as serotypes 20, 22, and 26 of *S. suis*, is rarely reported in clinics compared with *S. suis* because of the lack of appropriate detection methods that can be used to distinguish it from *S. suis* [18]. Given the lack of clinical isolates, the significance of *S. parasuis* for public health is underestimated [19]. The presence of *S. parasuis* has recently been reported in a few countries, such as China, Japan, Canada, and Switzerland [20,21,22,23]. The *S. parasuis* strain that harbors *optrA* and *cfr*(D) genes was first discovered in Qinghai Province, China, in 2018 [21]. Then, we accidentally isolated two *S. parasuis* strains that carry a chromosomal *optrA* gene and a plasmid-borne *cfr*(D) gene during drug-resistance monitoring on a pig farm in Guangdong Province, China, in 2021. Furthermore, the two *S. parasuis* isolates showed different sequence types. The strain has spread across a large geographic area among livestock and poultry. Here, we report two *S. parasuis* strains with *cfr*(D) and *optrA* from a Chinese pig farm.

## 2. Materials and Methods

### 2.1. Sample Collection and Bacterial Strains

A total of 912 samples (762 pig lung samples and 150 pig nasal swab samples) were collected from abattoir and pig farms in three provinces of China (i.e., Guangdong, Jiangxi, and Hunan) during 2020–2021. All samples were incubated in tryptic soy broth (TSB, 5% fetal bovine serum) and then streaked onto tryptic soy agar (TSA) with 5% defibrinated sheep blood. The rifampicin-resistant *E. faecalis* JH2-2 served as the recipient strain in the transfer experiments.

### 2.2. Antimicrobial Susceptibility Testing

Antimicrobial susceptibility testing for meropenem, rifampicin, linezolid, florfenicol, tetracycline, erythromycin, clindamycin, penicillin, ceftiofur, ampicillin, amoxicillin, enrofloxacin, cotrimoxazole, and vancomycin was performed using the broth microdilution method according to the guidelines of the Clinical and Laboratory Standards Institute (VET01-S2 and M100-S26) [24,25]. *Streptococcus pneumoniae* ATCC 49,619 was used as a quality control strain to determine the minimum inhibitory concentration.

### 2.3. PCR Analysis

The *optrA* and *cfr* genes were detected in all strains by using polymerase chain reaction (PCR), as described previously [26,27]. The presence of circular intermediates was detected by inverse PCR using the primers in-*optrA*-F, GGGAACAGTTGATGAGAGAA, and in-*optrA*-R, CCAACACCATATTACCATCAT (annealing temperature of 51 °C). Sanger sequencing was used for all PCR products.

### 2.4. Whole-Genome Sequencing (WGS) and Analysis

The genomic DNA of the *S. parasuis* strains SS17 and SS20 that carry both *optrA* and *cfr* genes was extracted using a HiPure Bacterial DNA Kit (Magen, Shanghai, China). The WGS was performed on an Illumina HiSeq TM2000 sequence platform (Novogene, Beijing, China) and PacBio RS Ⅲ sequencing platform (Tianjin Biochip Company, Tianjin, China). Draft genomes of Illumina HiSeq sequences were assembled using the CLC Genomics Workbench 10.0.1 (CLC Bio, Aarhus, Denmark) [28]. HGAP4 analysis was used to generate assemblies de novo and data statistics for the original genome data that was measured using Pacbio sequel technology [29]. The Rapid Annotation using the Subsystem Technology (RAST) (https://rast.nmpdr.org/, accessed on 7 August 2021) server was accessed for genome annotation. Antimicrobial resistance genes, virulence genes, and mobile genetic elements were identified using CGE ResFinder 3.2 (https://cge.cbs.dtu.dk/services/ResFinder/, accessed on 7 August 2021), Virulence Finder 2.0 (https://cge.cbs.dtu.dk/services/VirulenceFinder/accessed on 7 August 2021), and MGE (https://cge.cbs.dtu.dk/services/MobileElementFinder/, accessed on 7 August 2021), respectively. The genetic environments of *optrA* and *cfr* were analyzed using the BLAST program (http://blast.ncbi.nlm.nih.gov/Blast.cgi, accessed on 7 August 2021) and Easyfig 2.2.5 (developed by the Beatson Microbial Genomics Lab, Brisbane, Australia) [30].

### 2.5. Phylogenetic Analyses of S. parasuis Isolates

To construct the phylogeny of *S. parasuis*, all of the genomes of *S. parasuis* (*n* = 9, collected from NCBI) were extracted compared with the genomes of the *S. parasuis* isolates in this study. The phylogenic tree was constructed by RAxML, with the genome of SS20 used as a reference [31]. Snippy was employed to calculate the single nucleotide polymorphism (SNP) among the various genomes. The phylogenic tree was illustrated using iTOL [32].

### 2.6. Transfer Experiments

Conjugation experiments were performed by filter mating using rifampicin-resistant *E. faecalis* JH2-2 as the recipient strain and the two isolates, which carried the *cfr* gene on the plasmids and *optrA* in the chromosome, as donors [33]. Transconjugants were selected on brain–heart infusion agar plates containing 100 mg/L rifampicin and 32 mg/L florfenicol or 10 mg/L chloramphenicol, 20 mg/L chloramphenicol, and 30 mg/L chloramphenicol [1].

### 2.7. Nucleotide Sequence Accession Numbers

The complete genomes of *S. parasuis* SS17 and SS20 have been deposited in GenBank and assigned the nucleotide sequence accession numbers CP090522 and CP086728.

## 3. Results

### 3.1. Identification of cfr(D) and optrA in the Streptococcus Isolates

A total of 30 strains of *S. suis* (3.29%, 30/912) and 6 strains of *S. parasuis* (0.66%, 6/912) were isolated and identified from 912 samples. Among the pig lung samples, 26 *S. suis* isolates (3.41%, 26/762) and 4 *S. parasuis* isolates (0.52%, 4/762) were isolated from 762 samples, but none of these were positive for *optrA* or *cfr*(D). From the 150 pig nasal swab samples, 4 *S. suis* and 2 *S. parasuis* strains were isolated. The coexistence of *optrA* and *cfr*(D) in the two *S. parasuis* isolates, SS17 and SS20, was detected.

### 3.2. Antibiotic Resistance and Resistance Determinants

The resistance rate to erythromycin and clindamycin was over 90% among 36 Streptococcus strains, followed by tetracycline. Most strains remain susceptible to florfenicol, meropenem, and enrofloxacin. Antibiotic susceptibility tests showed that *S. parasuis* SS17 and SS20 demonstrated a multidrug resistance profile. They were resistant to florfenicol, erythromycin, clindamycin, tetracycline, penicillin, ampicillin, and enrofloxacin, but remained susceptible to meropenem, vancomycin, linezolid, amoxicillin, and ceftiofur. Acquired drug resistance gene test results showed that *S. parasuis* SS17 contained *ant*(6)-*la*, *optrA*, *aac*(6′)-*aph*(2″), *erm*(B), *tet*(M), and *cfr*(D), while *S. parasuis* SS20 contained *ant*(6)-*la*, *optrA*, *erm*(B), *tet*(M)*, msr*(D)*, mef*(A), and *cfr*(D) (Table 1).

### 3.3. WGS Analyses

The whole-genome sequence analysis showed that the chromosomes of SS17 and SS20 were 1,959,737 bp and 2,083,983 bp in size with GC contents of 39.6% and 39.5%, respectively. The genome of SS17 contained 2066 coding sequences and 46 RNA genes shown by RAST, while the SS20 contained 2275 coding sequences and 46 RNA genes. *S. parasuis* SS17 harbored a chromosomal *optrA* and a *cfr*(D)-carrying plasmid named pSS17. *S. parasuis* SS20 contained a chromosomal *optrA* and two plasmids named pSS20-1 and pSS20-2. The plasmid pSS20-1 carried a *cfr*(D) gene, while another plasmid, pSS20-1, was associated with no antimicrobial resistance gene.

### 3.4. Characterization of Plasmids Carrying cfr(D)

The plasmids pSS17 and pSS20-1, which carry *cfr*(D), had a 100% nucleotide sequence identity. They were 7550 bp in length with 37.8% GC content and have nine coding sequences and no RNAs. Except for the four open reading frames (ORFs) that encode hypothetical proteins, the remaining five ORF coding proteins were identified as *cfr*(D), GMP synthase, replication protein, IS1202, and IS431mec (Figure 1a,b). The nucleotide sequences of *cfr*(D) in pSS17 and pSS20-1 showed 100% (1074 of 1074) identity with the corresponding *cfr*(D) sequence from plasmid pH35-cfrD, which was from the *S. parasuis* strain H35 (GenBank accession no.CP076722.1) of porcine origin. The *cfr*(D) genes of pSS17 and pSS20-1 were flanked by GMP synthase, IS1202, and IS431mec. BLASTn analysis showed that pSS17 and pSS20-1 shared 99% (5334 of 5335) nucleotide sequence identity with the pH35-cfrD plasmid of the *S. parasuis* strain H35 isolated from a lung sample of a pig in Qinghai province, China, in 2018. In addition, SS17, SS20 and H35 had 100% identity in the *cfr*(D) gene sequences. This result indicated the possibility of a similar origin for pSS17, pSS20-1, and pH35-cfrD [21]. To identify the ability of the *cfr*(D) plasmid to conjugate, filter matings were performed using graded levels of chloramphenicol (10, 20, and 30 mg/L) or 32 mg/L florfenicol with the *E. faecalis* JH2-2 as the recipient strain. Nevertheless, no transconjugant was obtained in the triplicate assays.

### 3.5. Genetic Environment of optrA in the Chromosomal DNA

The *optrA* genes from SS17 and SS20 had 100% nucleotide sequence identity. Genetic environment analysis indicated that the *optrA* genes of SS17 and SS20 located on the chromosome were associated with the Tn*554* and *araC* genes. Tn*554* carried the resistance genes *erm*(A) and *ant*(9). The *araC* gene was a transcriptional regulator gene, forming a core segment of 3453 bp with *optrA*. However, the *optrA* of *S. parasuis* H35 was flanked by IS*1216E* elements. The *optrA* of SS17 and SS20 had three SNPs (791, T→C; 1120, G→A; 1729, T→C) compared with *S. parasuis* H35 from the same host in the same country. BLASTn analysis revealed that the *optrA*-carrying fragment exhibited high similarity with the corresponding region in the chromosomal DNA of the *E. faecium* strain GJA5 (GenBank accession no. MK251151.1), *Staphylococcus *sp. MZ7 (GenBank accession no. CP076027.1), *Staphylococcus* sp. MZ1 (GenBank accession no. CP076025.1), and the plasmid pL15 of the *E. faecalis* strain L15 (GenBank accession no. CP042214.1) (Figure 1c).

The transposon Tn*554* containing *erm*(A) and *ant*(9) was detected upstream of araC–optrA of SS17 and SS20, sharing 100% nucleotide sequence identity with the *E. faecalis* strain L15 plasmid (GenBank accession no. CP042214.1), the *E. faecium* strain GJA5, *Staphylococcus* sp. MZ7, and *Staphylococcus* sp. MZ1 (Figure 1c). Tn*554* may play an important role in the horizontal transmission of optrA in Gram-positive bacteria [34,35]. The araC–optrA gene clusters in SS17 and SS20 had 100% nucleotide sequence identity with the E. faecium strain GJA5, the E. faecalis strain L15 plasmid, and the S. parasuis strain H35 (Figure 1c). The ability of the araC- optrA gene clusters in E. faecalis to form circular intermediates and spread horizontally has been confirmed [1]. The ability of the gene to form a covalent closure circle appeared to enhance its ability to excise and integrate into the chromosome or other mobile genetic elements, as previously characterized [36,37,38]. Therefore, here, we used inverse PCR to determine the ability of optrA to form circular intermediates by using genomes of SS17 and SS20 as templates with the primers designed at both ends of the optrA gene. The result revealed that optrA formed a circle with its flanking 269 base pairs among the two strains. However, no transconjugant harboring optrA was obtained in the transfer experiments. It is still necessary to confirm the optrA’s transferability. 

### 3.6. Phylogenetic Relatedness of Streptococcus parasuis Strains

A phylogenic tree was produced to analyze the evolution of *S. parasuis*. The phylogenic tree demonstrated that all *S. parasuis* strains were closely related. No SNPs were identified in SS17 and SS20. Therefore, it is clear that S. parasuis has not evolved significantly among different hosts or geographical locations. (Figure 2). These strains carried only a few ARGs, e.g., *ant(6)-la* and *erm*(B), and some did not contain any ARGs. At the same time, the distribution of ARGs explained the results of the antibiotic resistance testing.

## 4. Discussion

A major factor in the prevalence of antibiotic resistance genes in the food chain is the presence of antibiotic-resistant bacteria in food animals, which poses a potential risk to public health and safety [39]. *Streptococcus suis* is an important zoonotic pathogen that can be transmitted to humans through contact with contaminated animal products or sick animals [40]. The ST25 and ST28 strains were dominant (9/30) among the isolates in this study. ST1 was identified as having a significantly higher virulence than ST25/28, and the remaining strains were ST242, ST27, ST1, ST7, and so on [41]. In addition to being extremely virulent, *S. suis* is recognized as a reservoir of ARGs, where transposons or integration and binding elements play a crucial role in the propagation of the organism [42]. *S. parasuis* is a close relative of *S. suis*, which may be an opportunistic pathogen, and it has been reported that it infects pigs, cattle, and humans [22]. The resistance of *S. parasuis* should also be taken into consideration. In this investigation, *S. suis* isolates only carried one to three ARGs, primarily *erm*(B) and *tet*(O), but *S. parasuis* isolates carried *ant(6)-la*, *aac(6′)-aph(2″)*, *mdt*(A), *lsa*(E), *cat*, *tet*(S) and other ARGs, which carry far more resistance genes. The detection rate of *tet*(O/M), *ant(6)-la*, and *erm*(B) in *S. parasuis* is higher, according to NCBI public data [43]. Based on these findings, *S. parasuis* is a potentially opportunistic zoonotic pathogen that may serve as a reservoir of resistance genes.

The spread of antibiotic resistance is significantly aided by both [44]. Since the *optrA* gene was discovered in China in 2015, it has been widely spread and detected in bacterial genera with diverse origins, including *Enterococcus* and *Staphylococcus*, in many different nations [45]. Platforms carrying the *optrA* gene can be categorized into three groups: those carrying *optrA* on chromosomally borne integration and conjugation elements (ICE), such as Tn*6674*, Tn*558*, and IS*1216E*; those on which the *optrA* gene is located on medium-sized plasmids (30–60 Kb) from the RepA_N, Inc18, and Rep_3 plasmid families to form *impB–fexA–optrA*; and those carrying *optrA–araC* on the chromosome or plasmids [46,47]. In this study, it was found that *optrA* was located in chromosomally borne Tn*554* carrying *erm*(A) and *ant*(9). The *cfr* gene was originally discovered on the multi-resistant pSCFS1 (16.5 kb) and pSCFS3 (35.7 kb) plasmids of *Staphylococcus sciuri* [48,49]. The *cfr*-carrying segment (IS*21-558-cfr*) was reported initially, which could form in tandem [50]. Mobile elements play an important role in the horizontal transfer of drug resistance genes. Subsequently, another multidrug-resistant plasmid (50 kb) carrying cfr was found in *Staphylococcus*, and a circular plasmid with five ORFs (*rep*-*Deltapre/mob*-*cfr*-*pre/mob*-*ermC*) was found in its transformants (7057 bp) [51]. Here, we found a new plasmid carrying the *cfr* gene in *S. parasuis*, 7550 bp, which is composed of 5 ORFs and *rep*-IS*431mec*-IS*1202*-*cfr*-*GME synthase*. Unfortunately, the transformants of *cfr* and *optrA* could not be obtained, and their transferability between different genera has yet to be confirmed. Interestingly, SS17 and SS20 remained sensitive to linezolid despite containing two oxazolidinone resistance mechanisms, *cfr*(D) and *optrA*. A study showed that *cfr*(D) did not produce any resistance when overexpressed in *E. faecalis* and *E. faecium*, but it was responsible for the phenicol resistance phenotype of *Escherichia coli* [52]. Meanwhile, the lack of *cfr* and *cfr*-like gene-mediated resistance to phenol and oxazolidinone in *E. faecalis* and *E. faecium* has been much reported [53,54]. Perhaps this phenomenon does not only occur in a single species. In contrast to the ubiquity of resistance in Gram-negative bacteria, Gram-positive bacteria still seem to preserve a defense against drug resistance genes. Since *optrA* was first identified, at least 69 variants have been identified; it is inferred that there are 1 to 20 amino acid differences. The different *optrA* variants may have an effect on the MIC of the oxazolidine of the corresponding isolates that show sensitivity/resistance [7].

In conclusion, this study reported the co-occurrence of *optrA* and *cfr*(D) operons in *S.parasuis*. The presence of Tn*554–optrA* in *Enterococcus* spp., *Staphylococcus* spp., and *Streptococcus* spp. with various genetic and source backgrounds demonstrated that the Tn*554* element plays an important role in the dissemination of *optrA*. Attention should be paid to the potential risks of plasmid-borne *cfr*(D) transference from streptococcus to other Gram-positive bacteria. Meanwhile, the existence of additional resistance genes, *erm*(A) and *ant*(9), and the use of various types of antibiotics may contribute to the prevalence of *optrA* and *cfr*(D). Therefore, it is urgently necessary to continuously monitor the spread of *optrA* and *cfr*(D) among Gram-positive bacteria, and monitor the prudent use of antibiotics in food animals.

## Figures and Tables

**Figure 1 antibiotics-12-00825-f001:**
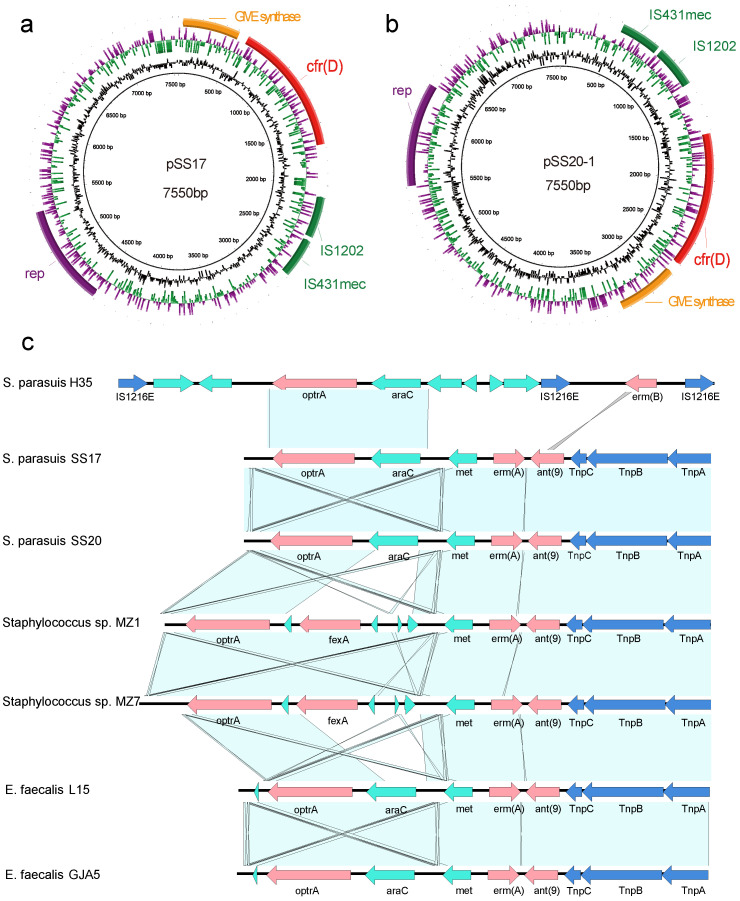
(**a**,**b**) are the structure and organization of the plasmids of pSS17 and pSS20-1, respectively. The circles (from the outside to inside) indicate the predicated coding sequences, GC-skew [(G + C)/(G + C)], GC content and scale in bp. The coding sequences with different functions are shown in different colors. Arrows indicate the direction of transcription of the genes. (**c**) Genetic environment of *optrA* in the chromosomal DNA of *S.parasuis* SS17 and SS20 compared to other plasmids and genomes. Resistance genes are indicated with pink arrows. Transposases are shown with blue arrows labeled by their name. Other elements are highlighted with green arrows. Shared regions with >99% identity are denoted by nattier blue shading.

**Figure 2 antibiotics-12-00825-f002:**
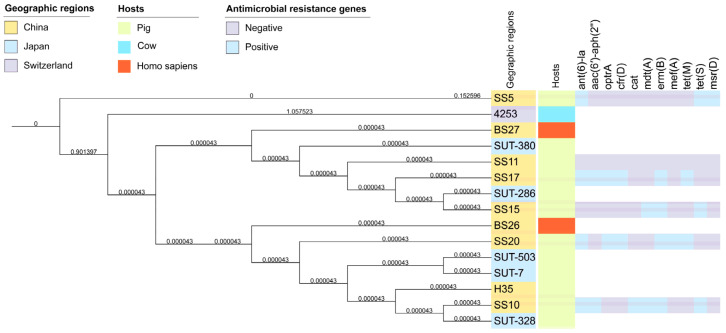
The phylogenetic analysis of *S. parasuis* isolates. The geographic regions of the sources of these isolates and the hosts are displayed in different colors. The antimicrobial resistance genes are indicated using a heatmap; light blue is positive, and purple is negative.

**Table 1 antibiotics-12-00825-t001:** Minimum inhibitory concentrations of 13 antimicrobial agents and ARGs among 2 *Streptococcus parasuis* isolates carrying both *optrA* and a *cfr*(D).

Isolate	Species	Source	MLST	MIC (mg/L)													
				ERY	CLI	MEM	CEF	LZD	VAN	TET	PEN	AMP	ENR	SXT	FFC	AMO	Resistance Genes
SS17	*S. parasuis*	nasal swab	NA	32	>32	≤0.03	0.5	1	0.12	64	2	1	4	32	64	0.5	*ant*(6)*-la*, *aac*(6′)-*aph*(2″), *optrA*, *erm*(B), *tet*(M), *cfr*(D)
SS20	*S. parasuis*	nasal swab	NA	32	>32	≤0.03	0.5	1	0.12	64	2	1	4	64	>64	1	*erm*(B), *ant*(6)-*la*, *optrA*, *mef*(A), *tet*(M), *cfr*(D)

MLST, multi-locus sequence type; MIC, minimum inhibitory concentration; ERY, erythromycin; CLI, clindamycin; MEM, meropenem; CEF, ceftiofur; LZD, linezolid; VAN, vancomycin; TET, tetracycline; PEN, penicillin; AMP, ampicillin; ENR, enrofloxacin; SXT, cotrimoxazole; FFC, florfenicol; AMO, amoxicillin; NA, not available.

## Data Availability

The data presented in this study are available in article.

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
