# Peer review of "Co-Existence of Oxazolidinone Resistance Genes cfr(D) and optrA on Two Streptococcus parasuis Isolates from Swine"

_antibiotics, 2023, doi:10.3390/antibiotics12050825_

Round 1
Reviewer 1 Report
The manuscript of Han et al. described their study on oxazolidone resistance screen and its molecular analysis of S. parasuis/suis isolates. The main emphasis was put on two MDR strains for whom WGS was performed and the data were analysed exhaustively. They did some mobilty studies too. Overall the study yielded some srceening reults, extensive and interesing and confirmatory molecular analysis and the mobilities of the important cfr(D) and optrA genes were also assessed.
Special comments
Line 236: The ability for transfer of the optrA gene... would be better to write
Line 240: Genera in italics
Line 242: optrA in italics
Line 249: 'Mobile' would be better
Lines 259-260: Escherichia coli in italics too, 'phenicol' antibiotics were used
Reviewer 2 Report
In this study the authors find oxazolidinone resistance genes cfr(D) and optrA on two Streptococcus parasuis isolates from swine, it's an important discovery and will be interest to the future treatment of antibiotics resistant strain. Overall is a good project, but I found the authors need to work on the language and grammar a little bit. I found some errors listed as follows:
1. Line 15 please delete Objective:
2. Line 16 please delete Methods:
3. Line 22 please delete Resutls:
4. Line 26 please delete Conclusions:
5. Line 18 " with screened using PCR" , please delete with screened
6. Line 22 namely, SS17 and SS20, please change to " named SS17 and SS20 respectively"
7. Line 65 move "2.1 Sample collection and bacterial strain to a separate line
8. Line 165, 198,199,200, 203, please unbold Gene accession number.
Reviewer 3 Report
Thank you for the opportunity to review this manuscript. I found the content and theory interesting. Nevertheless, there are some issues with the manuscript that must be improved. My specific comments are below:
General comments:
I think the conclusions of this manuscript are being overstated. My specific comments are below but I'm not sure that the conclusions being made are valid within the context that they are being presented. While this study has merit, I'm not sure it's publishable in it's current form without significant changes
Introduction
It seems as though that an argument is being made that these pathogens have public health implications. That being said, no evidence is presented in the introduction to support this assertion. What is being presented is vague and leaves the reader to assume or infer. This must be addressed.
lines 37-39: In what population are these antimicrobials being used on a widespread basis?
Lines 41 and 42: Please define what "considerable" bacterial species means
Line 46: Should this be phenicols?
Lines 53-54: What clinics are being discussed here?
Materials and Methods
Lines 66-71: More description on the animals from which these isolates were obtained is absolutely necessary. Why did the animals from which lungs samples were collected die? Why were nasal swabs collected? What was the history of antimicrobial use on the farm and in these animals, specifically? These are obviously biased samples since they are collected from a non-random subset. How does this apply to the prevalence in the larger herd?
Lines 77-78: Current CLSI guidelines (VET01S) recommend the use of S. pneumoniae ATCC 49619 as the QC strain. This somewhat invalidates the data being presented
Results
Lines 119-124: Only 36 isolates were collected from more than 900 samples. This is a relatively low prevalence, especially since these samples were non-randomly selected. Moreover, only 2 were positive for the genes of interest. While the finding is interesting, it really doesn't suggest a significant issue. Again, without context of antimicrobial use this is even harder to interpret.
In line 122, the sentence should never begin with a number.
Table 1: MIC's for two isolates don't really give much information. More isolates are needed (general recommendations from CLSI suggest 30 or more). Moreover, they are susceptible to linezolid despite carriage of the genes of interest. I would suggest presenting the MIC data for the S. suis isolates and providing the reader with more context on animal history, farm managment, and antimicrobial use patterns.
Figure 1: Completely unreadable. Quality is very poor.
Discussion
Lines 227-228: Really no evidence for this statement from this study.
Lines 235-237: Overstatement of the issue. Only 2 isolates from nearly 1000 samples. This conclusion can't be drawn.
Round 2
Reviewer 3 Report
All of my previous concerns have been addressed